# Dissociable consequences of moderate and high volume stress are mediated by the differential energetic demands of stress

**Michael A. Conoscenti**[1,2]*, **Nancy J. Smith**[1,2], **Michael S. Fanselow**[1,2,3]

1 Department of Psychology, University of California, Los Angeles, Los Angeles, California, United States of America, 2 Staglin Center for Brain & Behavioral Health, University of California, Los Angeles, Los Angeles, California, United States of America, 3 Department of Psychiatry & Biobehavioral Sciences, University of California, Los Angeles, Los Angeles, California, United States of America

* michael.conoscenti@neuro.utah.edu

**Data Availability Statement:** A PDF of uncropped Western Blot images has been added to the Github repository that included all raw western blot files. Again the links to data and blots are as follows:

## Abstract

Exposure to traumatic stress leads to persistent, deleterious behavioral and biological changes in both human and non-human species. The effects of stress are not always consistent, however, as exposure to different stressors often leads to heterogeneous effects. The intensity of the stressor may be a key factor in determining the consequences of stress. While it is difficult to quantify intensity for many stress types, electric shock exposure provides us with a stressor that has quantifiable parameters (presentation length x intensity x number = shock volume). Therefore, to test the procedural differences in shock volume that may account for some reported heterogeneity, we used two common shock procedures. Learned helplessness is a commonly reported behavioral outcome, highlighted by a deficit in subsequent shuttle-box escape, which requires a relatively high-volume stress (HVS) of about 100 uncontrollable shocks. Conversely, stress-enhanced fear learning (SEFL) is another common behavioral outcome that requires a relatively moderate-volume stress (MVS) of only 15 shocks. We exposed rats to HVS, MVS, or no stress (NS) and examined the effects on subsequent fear learning and normal weight gain. We found doubly dissociable effects of the two levels of stress. MVS enhanced contextual fear learning but did not impact weight, while HVS produced the opposite pattern. In other words, more stress does not simply lead to greater impairment. We then tested the hypothesis that the different stress-induced sequalae arouse from an energetic challenge imposed on the hippocampus by HVS but not MVS. HVS rats that consumed a glucose solution did exhibit SEFL. Furthermore, rats exposed to MVS and glucoprivated during single-trial context conditioning did not exhibit SEFL. Consistent with the hypothesis that the inability of HVS to enhance fear learning is because of an impact on the hippocampus, HVS did enhance hippocampus-independent auditory fear learning. Finally, we provide evidence that stressors of different volumes produce dissociable changes in glutamate receptor proteins in the basolateral amygdala (BLA) and dorsal hippocampus (DH). The data indicate that while the intensity of stress is a critical determinant of stress-induced phenotypes that effect is nonlinear.

https://github.com/mconoscenti/shock_volume and https://github.com/mconoscenti/shock_volume/tree/mconoscenti-westernblot-raw.

**Funding:** This research was supported by NIH Grant R01-MH115678 (M.S.F.) and funds from the Staglin Center for Brain & Behavioral Health (M.S.F.).

**Competing interests:** The authors have declared that no competing interests exist.

## 1. Introduction

Acute, intense stressors can lead to various physiological and psychological conditions in both human and non-human species [1, 2]. Post-Traumatic Stress Disorder (PTSD) develops in about 20% of those that experience a traumatic stressor [3]. PTSD is a debilitating and heterogeneous disease marked by a wide variety of potential symptoms such as amnesia, anhedonia, avoidance behaviors, exaggerated fear-potentiated startle, hypervigilance, and insomnia [4]. PTSD patients also exhibit a diverse array of comorbidities, including major depressive disorder and substance use disorders [5–8]. Great strides have been made in understanding the neurobiological consequences of severe stress, yet little headway has been made in identifying effective treatment(s) of stress-induced psychiatric diseases such as PTSD.

One apparent roadblock to advancing basic research on stress and its translation to clinically effective treatment is the highly inconsistent findings among research groups. Inconsistent findings may be because stress models vary widely between research groups, leading to divergent behavioral and biological findings [9, 10]. Despite the disparity among groups, there has been little to no attempt to consolidate the stress literature. The many qualitatively different stressors used in the laboratory makes it nearly impossible to compare findings. However, electric shock exposure provides a stressor in which the parameters are quantifiable (duration, intensity, and number of presentations). There is prior evidence that the amount of shock acts as a determining factor for qualitative differences in stress-reactive behavior [11]. In a recent review, we looked at two comparable stress procedures due to their mutual use of inescapable and unpredictable electric shock as the stressor [9]. This review compared the behavioral and biological impacts of the stressors used to induce two often explored behavioral effects, *learned helplessness* and *stress-enhanced fear learning.*

The stressor that produces the *learned helplessness* (LH) effect typically consists of 100, 1 mA tail shocks of variable length (based on the yoked controls performance, this typically averages between 3–8 seconds over the course of the session) that occur during a 2-hour session [12, 13]. The hallmark behavior is the subsequent deficit in escape performance within the shuttle-box apparatus [14, 15]. However, rats exposed to the extensive 100 tailshock session also exhibit a wide array of behavioral characteristics that parallel several symptoms of PTSD and depression [16, 17]; for review, see [9]. For example, rats that are exposed to 100 tailshocks show anhedonia and lower weight than controls [18–21]. Several biological mediators of the shuttle-escape deficit have been identified, such as corticosterone, serotonin, and adenosine [12, 22, 23]. The impact of the energetically demanding fear state caused by this extensive stress session is also implicated in the deleterious behavioral consequences of stress. The 100-shock session transiently stresses energy homeostasis [24]. Access to a concentrated glucose solution reverses the shuttle-escape deficits produced by the stressor [25, 26], while artificial glucoprivation using 2-deoxy-D-glucose (2-DG) and adenosine agonists promote shuttle-escape deficits in unstressed rats [27, 28]. Furthermore, the activity of adenosine, which is closely linked to cellular energy homeostasis, is both necessary and sufficient for the stress-induced shuttle escape deficits [23, 27, 29, 30]. These findings led to our previous hypothesis that the energetic challenge induced by the stressor is a key mediator for the observed deleterious behavioral effects.

The *stress-enhanced fear learning* (SEFL) stressor consists of 15, 1 mA footshocks of fixed length (1 second) occurring during a 1.5-hour session. This stress procedure and subsequent behavioral phenomena became a focus of interest due to its ability to produce a long-lasting (e.g., at least 3 months) enhancement of subsequent contextual and auditory fear conditioning [31–33]. It was later discovered that this shock procedure also produced a robust array of anxiety-like behaviors [34, 35]. Rats exposed to 15 shocks also exhibit a wide array of behavioral

characteristics similar to the symptoms of PTSD but do not exhibit depression-like behavior as reported following 100 shocks [9, 35]. Evidence of the neurobiological mediators for SEFL is relatively limited, but initial evidence points toward a rise in basolateral amygdala (BLA) GluA1, which forms a functionally unique tetramer of the α-amino-3-hydroxy-5-methyl-4-isoxazolepropionic acid (AMPA) receptor, in the BLA, as a mediator for the sensitization effect [35].

The stressors that induce LH and SEFL produce several similar behavioral characteristics that model pathological fear and anxiety in the rat. The research also suggests that they may diverge in inducing depression-like behavior. However, research has not directly compared the LH and SEFL inducing stressors. Here we test the hypothesis that more extensive exposure to a stressor equates to greater behavioral and biological consequences. The LH and SEFL producing stressors are particularly useful in testing this because they are qualitatively similar yet vary on a major dimension: shock volume (shock number x current x length). Rats exposed to an LH-inducing stressor experience a total of about 300 to 800 seconds of 1 mA shock (100 shocks at an average of 3–8 seconds each, based on yoked performance), while SEFL-exposed rats receive a total of only 15 seconds of 1 mA shock (15 shocks at an average of 1 second each). To avoid confusion between the stressors and their behavioral consequences, and make it clear that we are not examining the effects of stressor controllability that have already been well characterized [12], we will subsequently refer to the 100-shock procedure as *high-volume stress (HVS)*, the 15-shock procedure as *moderate-volume stress (MVS)*, and the restraint controls as *no shock (NS)*. While it is clear that both MVS and HVS produce similar levels of anxiety-like behavior in the animal, and only HVS produces consistent depressive-like behavior, their individual effects on fear have yet to be comprehensively examined. Here we test the hypothesis that enhanced fear learning is differentially expressed following MVS and HVS exposure.

Six experiments investigate the behavioral and neurobiological consequences of stress volume. Rats were restrained in tubes and exposed to either 0, 15, or 800 cumulative seconds of shock over a 90 or 113-minute interval. Rats were assessed for enhanced fear learning or sacrificed for tissue analysis one week after stress pretreatment. We included manipulations that increase or decrease glucose availability, with the hypothesis that glucose availability only mediates effects produced by the energetically demanding high-volume stressor. These manipulations either occurred immediately after the termination of the stress session or immediately before single-trial conditioning. Rats were weighed throughout the study, and weight gain was compared across groups.

## 2. Materials and methods

### 2.1. Subjects

Two hundred and eight Sprague-Dawley albino male rats (290–320 grams) from Envigo (Placentia, CA, USA) were housed in individual cages in a room maintained on a 12:12-hour light/dark cycle (6:00–17:59 lights on, 18:00–5:59 lights off). Animals were housed in the room for approximately two weeks prior to testing. Rats were housed in metal hanging cages. Each cage was equipped with a standard glass (250 mL) water bottle, with a rubber stopper and metal spout, and a metal food hopper that allowed to *ad libitum* consumption of water and standard rat chow. During this time, all animals had free access to food and water. All experimentation occurred during the early light cycle (7:00–10:00, approximately). The protocols in this paper received pre-approval by the UCLA Institutional Animal Care and Use Committee.

## 2.2. Apparatus

Rats were restrained in clear Plexiglass restraining tubes during stress pretreatment, as previously described [25]. Electric shock was administered via electrodes attached to a rat's extended tail. During the session, each restraining tube was housed in an illuminated, sound-attenuating chamber. Testing occurred in Med Associates (St Alban, Vt) behavioral testing chambers. Each chamber was equipped with an infrared camera, speaker for tone delivery, shock scrambler, and fluorescent and infrared light sources. The behavioral testing chambers in each testing room were controlled by a PC using Med Associates Video Freeze software that automatically scored the animal's shock-induced motion and freezing during the test session. Modification of the chamber's contextual features used differential lighting, odors, ambient noise, and interchangeable grid floors and wall inserts to create distinct contexts when necessary.

## 2.3. Procedure

Rats were assigned randomly to groups of eight to ten rats each. Rats received restraint, fifteen (MVS) or one hundred (HVS) inescapable tailshocks. One day or one week later, rats underwent a fear conditioning procedure or were sacrificed for tissue analysis.

Rats that received HVS were exposed to 100, 1.0 mA variable-duration (mean = 8.0 s, range: 3 to 15 s) and inescapable tail shocks on a variable-time 60-s schedule (range: 20 to 150 s) in restraining tubes during a 113-min stress pretreatment session. Rats that received MVS shock were exposed to 15, 1.0 mA fixed-duration (1 second) and inescapable tail shocks on a variable-time 360-s schedule (range: 120 to 900 s) in restraining tubes during a 90-min stress pretreatment session. The specific HVS and MVS parameters were chosen to mimic previously published work that produced stress-induced shuttle-escape deficits [36] or stress-enhanced fear learning [31]. However, prior research with the MVS SEFL procedure always used footshock while prior work with HVS LH primarily used tailshock. To make the procedures more comparable we used tailshock for both our MVS and HVS stressors. The other groups were restrained in tubes for the same period (113 or 90 minutes) and received no shock. A home cage control (HCC) was added for all experiments involving tissue analysis. These animals were handled the same as other groups but were not exposed to the stress pretreatment.

The fear conditioning procedure was as follows. On the first day of testing, rats were placed in a novel environment and received a single, 1-second and 1 mA footshock after three minutes of free exploration. Rats were retrieved thirty seconds after shock exposure and returned to their home cage. The following day, rats were placed back into this context for eight minutes to test for contextual fear. Time spent freezing was assessed during both days. In the cued fear learning experiment, a 30 second, 65 dB, 2800 Hz tone preceded and co-terminated with shock. All rats were tested for contextual fear conditioning in the cued learning experiment as previously stated. All groups were then pre-exposed to a novel context one day after contextual fear conditioning testing. Preexposure consisted of three, 30-minute sessions across three days (1 session/day). Following preexposure, all rats received a tone test where, following a three-minute baseline period, three, 30-second tone presentations were spaced one minute apart.

In the experiments involving glucose manipulation, all stress and contextual fear conditioning procedures were identical to those previously described, but with the addition of glucose or 2-DG delivery. In the glucose intervention experiment, all rats were pre-exposed to a glucose cocktail over three consecutive days [26]. During glucose preexposure, each animal's water bottle was replaced with a bottle containing the glucose cocktail. Glucose preexposure occurred 10 days prior to stress exposure. The cocktail consisted of 40% glucose and 5% sucrose dissolved in tap water (weight/volume). Rats received 18-hours of free access to

glucose or water, based on experimental condition, immediately following the termination of stress pretreatment. In the experiment involving peripheral injection of 2-DG, rats were injected intraperitoneally with either vehicle or 100 mg/kg of 2-DG dissolved in distilled water twenty minutes before single-trial fear conditioning.

## 2.4. Western blot analyses

The dorsal hippocampus (DH), ventral hippocampus (VH), and BLA were dissected and flash frozen for western blot analysis. Tissue was homogenized and spun to separate crude and synapto-neurosome homogenate and diluted in a synaptic protein extraction reagent containing protease and phosphatase inhibitors (ThermoFisher, Cat #s 87793 & 78440). Protein concentrations of diluted homogenate were estimated using BCA assay (ThermoFisher, Cat # 23225). 15ug of protein was loaded into a 10% polyacrylamide gel for electrophoretic separation and then transferred to a PVDF membrane (Bio-Rad, Cat #s 5671035 & 1704157). Lanes were assessed for total protein using Ruby protein blot staining (ThermoFisher, Cat # S11791). Primary antibody was then applied overnight and secondary antibody (fluorescent or chemiluminescent) was applied for one to two hours the following day. Tissue was analyzed for GluA1 (Millipore cat # ABN241, 1:5000), GluA2 (Millipore cat # MABN1189, 1:1000), NR1 (Millipore cat # AB9864, 1:5000), NR2a (Millipore cat # AB1555P, 1:10000), NR2b (Abcam cat # AB28373, 1:5000), and GAPDH (Abcam cat # AB8245, 1:5000). Secondary antibodies were applied at a 1:10000 to 1:5000 dilution depending on primary antibody specifications (Abcam cat # AB205719, Bio-Rad cat #s 12005867 & 12004162). Blots were imaged using a ChemiDoc MP imager and analyzed using Image Lab software (Bio-Rad, cat #s 17001402 & 1709690).

## 2.5. Statistical analysis

Software package SPSS (SAS Institute, Inc., Version 16.0, Cary, NC, USA) was used for statistical analyses. One-way, two-way, three-way, and mixed-design ANOVAs were used when appropriate. Following significant interactions, Tukey post-hoc analyses were reported. Statistical significance was noted when p values were less than 0.05. Data are presented as individual data points overlaid by group means with error bars denoting group mean +/− SEM. No statistical outliers were removed from the data. Animals were excluded solely based on equipment malfunction.

## 3. Results

### 3.1 Moderate (MVS) and high-volume (HVS) stressors result in dissociable behavioral effects

We tested the hypothesis that rats exposed to HVS and MVS will exhibit distinct behavioral phenotypes. Due to the greater volume of shock exposure present in HVS, we initially hypothesized that rats exposed to HVS would exhibit heightened SEFL and greater persistent weight loss compared to rats exposed to MVS.

Fig 1 shows percent freezing to the conditioning context (A & C) and weight change (B & D) one day and one week following stress pretreatment. The MVS group showed higher levels of contextual fear-induced freezing compared to the NS and HVS groups during the test, regardless of the time that intervened between stress exposure and fear conditioning (A & C). One-way analyses of variance (ANOVA) on freezing during the context test yielded a significant main effect of group on freezing (%) both when the interval between stress exposure and fear conditioning was one day or one week, $F_{(2, 20)} = 8.010$, $p = .003$ and $F_{(2,20)} = 7.095$, $p = .005$. Tukey post-hoc comparisons on group means indicated a relationship among groups:

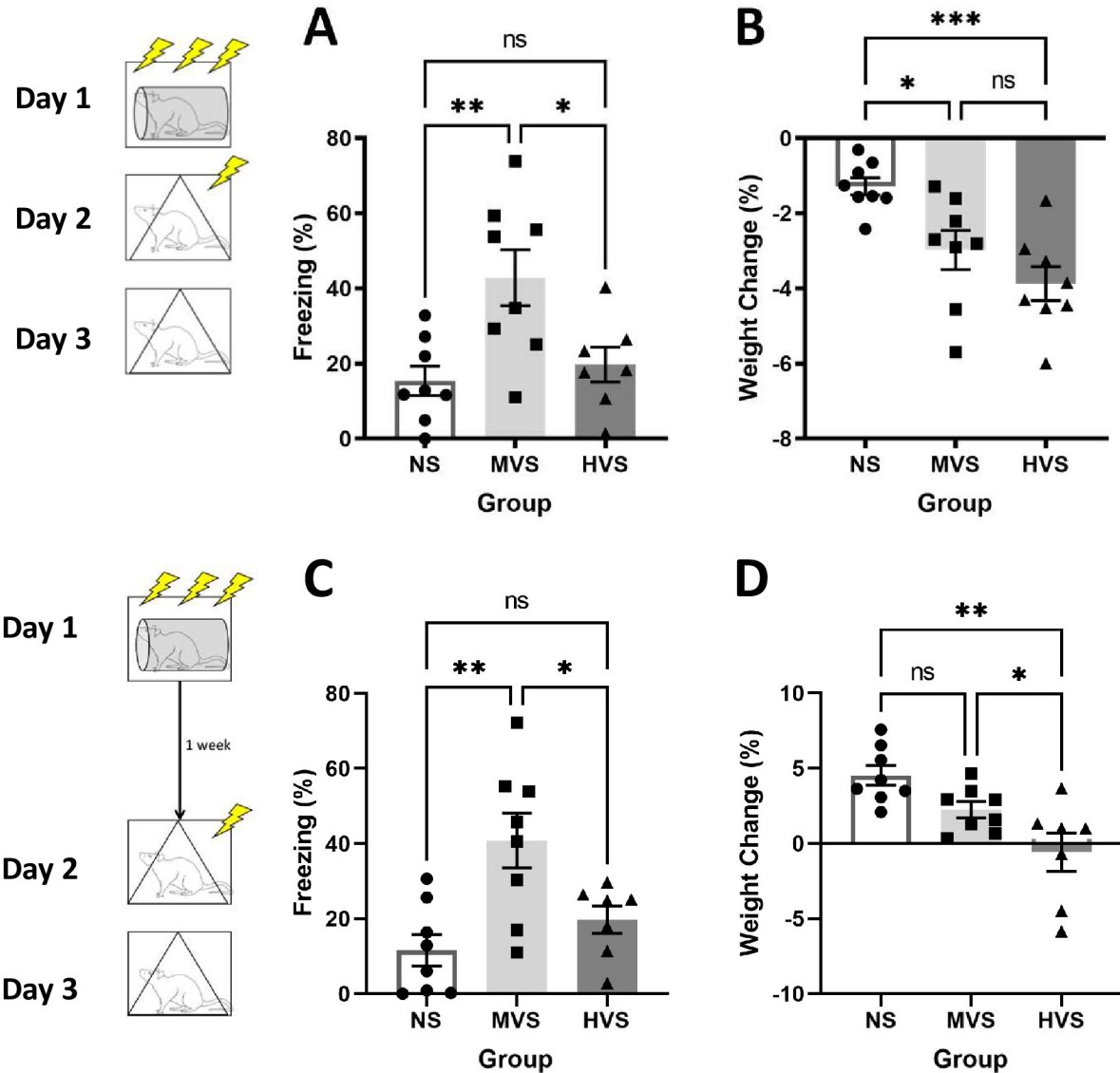

**Fig 1. Effects of stress volume on fear learning and weight maintenance.** Depicted: Percent freezing during context fear test one day (panel A) and one week (panel C) following stress exposure; weight change one day (panel B) and one week (panel D) following stress exposure. Rats were exposed to 0 (NS), 15 (MVS), or 100 (HVS) tailshocks one day or one week prior to single-trial fear conditioning. Testing consisted of exposure to a single, 1 mA shock in a novel context. Rats were then returned to this same context 24-hours later. Rats were weighed prior to stress exposure and prior to fear conditioning testing. The MVS group spent more time freezing during the context group when compared to NS and HVS groups. The HVS group showed significantly less weight gain when compared to MVS and NS groups one week after stress exposure. Both shock groups exhibited greater weight loss, compared to NS, one day following stress exposure. Error bars denote mean ± SEM. * p < .05, ** p < .01.

MVS > HVS = NS. Both stress groups exhibited greater weight loss compared to NS 24-hours following stress exposure (B). A one-way ANOVA on Weight yielded a significant main effect of Group, $F_{(2, 20)} = 9.729$, p = .001. Tukey post-hoc comparisons on groups means indicated a relationship among groups, such that: MVS = HVS < NS. Interestingly, the HVS group showed diminished normal weight gain one week following stress exposure compared to MVS and NS groups (D). A one-way ANOVA on Weight yielded a significant main effect of Group,

F (2, 20) = 8.860, p = .002. Tukey post-hoc comparisons on group means indicated a relationship among groups, such that: MVS = NS > HVS.

The rats exhibited no freezing prior to the single shock exposure. Therefore, there were no significant differences in baseline freezing (F < 1). This suggests that the differences observed in freezing during the context test are indeed due to differences in fear learning, not generalization between the stress and fear conditioning contexts. Furthermore, no significant differences in shock reactivity were observed as measured by the average motion index (F < 1).

## 3.2 Moderate and high-volume stress exposure results in enhanced cued fear conditioning

The previous experiment showed that differences in stress volume impact subsequent behavior and physiology of the animal. Specifically, when one week intervened between stress exposure and fear conditioning, we found a double-dissociation such that rats exposed to MVS exhibited SEFL but did not have suppressed weight gain; rats exposed to HVS showed the opposite pattern. HVS has a great physiological impact on the hippocampus [37, 38], but evidence of the stressor's functional impact remains elusive [39]. While there is no evidence of HVS's impact on hippocampal function during an unstressed state, a small body of evidence suggests that hippocampal processing may be impaired during any subsequent testing that also elicits the stress response [40, 41]. Contextual fear conditioning is reliant upon hippocampal processing [42, 43]. However, conditioning to discrete stimuli, such as a tone, typically does not rely on the hippocampus [44–47]. Here, we attempt to test the hypothesis that HVS impairs the enhancement of subsequent contextual fear conditioning by decreasing hippocampal function during a stressful event. Specifically, we hypothesize that while rats exposed to HVS do not express SEFL to a context, they will express SEFL to a tone- an association that does not require the hippocampus [48].

Fig 2 shows percent freezing to the 1-shock context (A), to the shock-associated tone (B), and during preexposure to a novel context (C). The MVS group showed higher freezing levels than the NS and HVS groups during the contextual fear test (A). A one-way ANOVA on freezing during the context test yielded a significant main effect of Group, F (2, 24) = 7.944, p = .0023. Tukey post-hoc comparisons on group means indicated a relationship among groups: MVS > HVS = NS. However, compared to the NS group, both MVS and HVS groups exhibited higher levels of freezing to the tone (B). A one-way ANOVA on freezing during tone presentation yielded a significant main effect of Group, F (2, 22) = 4.327, p = 0.0260. Tukey post-hoc comparisons on group means indicated a relationship among groups: NS < MVS = HVS. A mixed-design ANOVA on freezing during context preexposure (C) yielded a significant Group x Trial interaction, F (4, 26) = 3.185, p = .0296. However, Tukey's post-hoc comparisons on group means did not indicate any statistically significant simple main effects. No differences in baseline freezing prior to the single shock exposure or tone presentation were observed (Fs<1).

## 3.3 Post-stress glucose rescues contextual SEFL behavior in rats exposed to high-volume stress

The previous experiment showed that while rats exposed to HVS did not exhibit enhancement of hippocampal-dependent contextual fear learning, they did express increased hippocampal-independent cued fear conditioning. Prior studies have indicated that glucose ingestion following HVS reverses several of the stressor's behavioral impacts [24–26]. It has been hypothesized that hippocampal-encoding of the context may mediate glucose's prophylactic effect [25, 40, 41]. We therefore hypothesized that post-stress glucose may induce the SEFL phenotype

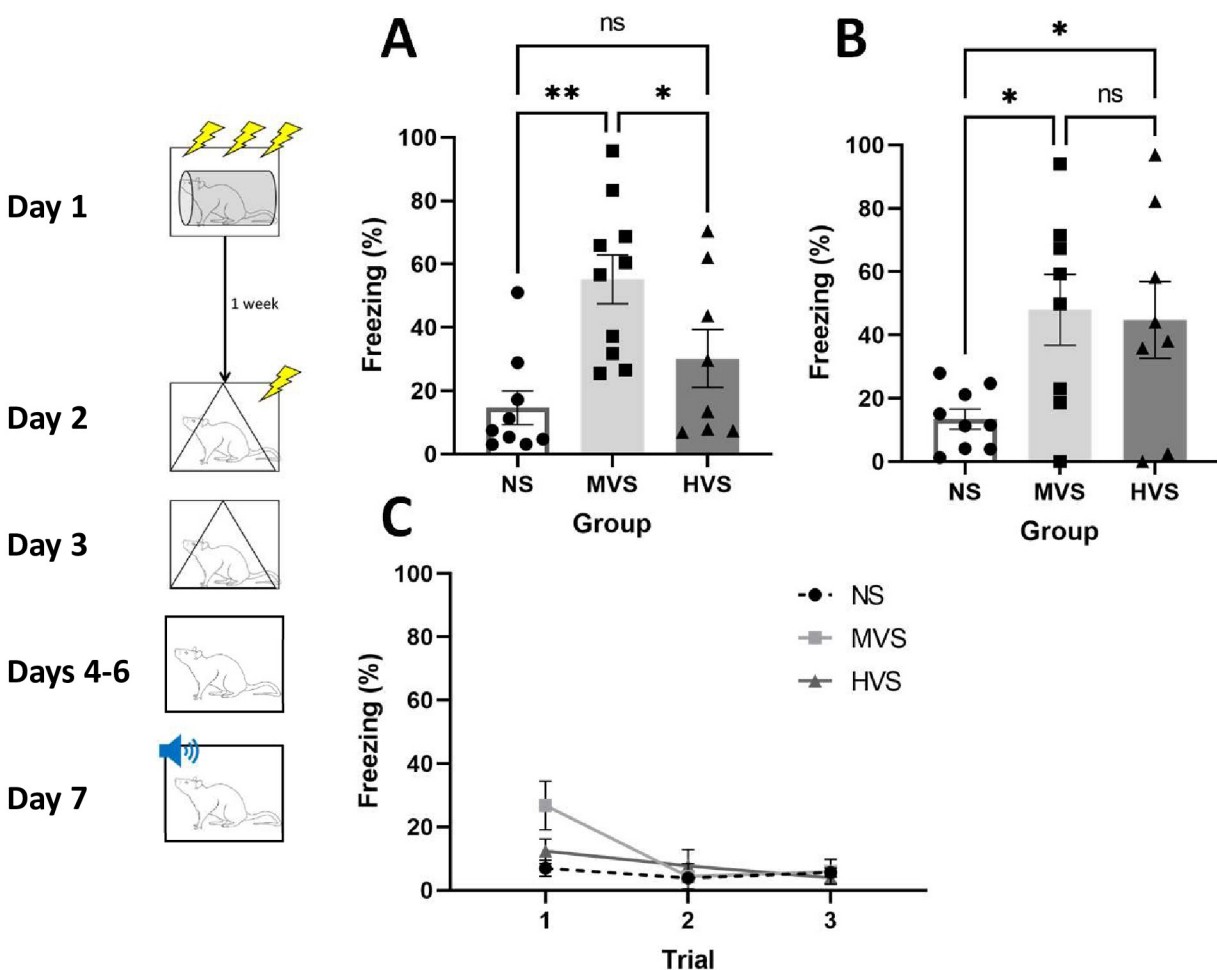

**Fig 2. Moderate and HVS exposure results in enhanced cued fear conditioning.** Depicted: Percent freezing during context conditioning test (panel A), presentation of the conditioned tone (panel B), and the first eight minutes of each context preexposure trial (panel C). Rats were exposed to 0 (NS), 15 (MVS), or 100 (HVS) tailshocks one week prior to single-trial fear conditioning. Testing consisted of exposure to a single, 1 mA shock in a novel context following presentation of a 30-second tone. Rats were then returned to this same context 24-hours later. Rats were then pre-exposed to another novel context. In this context, the previously-conditioned tone was presented and freezing was assessed. The MVS group spent more time freezing during the context test when compared to NS and HVS groups. Interestingly, both MVS and HVS groups showed higher levels of freezing to the tone compared to the NS group. Error bars denote mean ± SEM. * p < .05, ** p < .01.

not previously observed in rats exposed to HVS. We also hypothesized that glucose may mitigate the suppression of weight gain observed following HVS.

Fig 3 shows percent freezing to the conditioned context (A) and weight change (B) one week following stress pretreatment. In rats given water following shock, the MVS group showed higher levels of freezing compared to the NS and HVS groups during the context test (as seen in the previous experiments). However, rats given glucose following HVS exhibited freezing levels higher than their water-drinking counterparts and similar to rats given MVS (A). A two-way ANOVA on freezing during the context test yielded a significant Stress x Fluid interaction, $F(2, 42) = 3.499$, $p = .0393$. Tukey post-hoc comparisons on group means indicated a relationship among groups: NS-W = NS-G = HVS-W < HVS-G = MVS-W = MVS-G. Weight gain was depressed in both HVS and MVS groups compared to NS. Furthermore, there appeared to be an overall depression of weight gain in groups that received access to post-stress glucose (B). A two-way ANOVA on weight change (%) yielded significant main

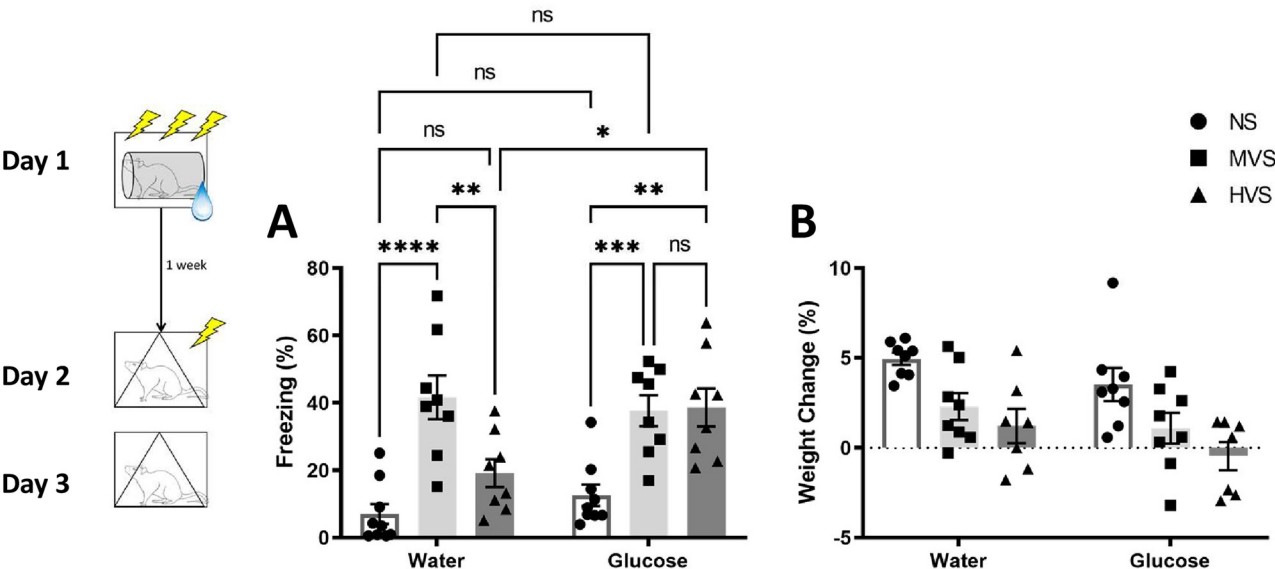

**Fig 3. Impacts of glucose ingestion on the fear learning and weight maintenance stress volume effects.** Depicted: Percent freezing during context conditioning test (panel A) and weight change one week following stress exposure (panel B). Rats were exposed to 0 (NS), 15 (MVS), or 100 (HVS) tail shocks one week prior to fear conditioning testing. Following stress exposure, all groups received 18-hour free access to a 40% glucose solution or tap water, based on experimental condition. All bottles were then switched back to tap water for the remainder of the experiment. Testing consisted of exposure to a single, 1 mA shock in a novel context. Rats were then returned to this same context 24-hours later. Rats were weighed prior to stress exposure and prior to fear conditioning testing. In groups that received water only, the MVS group spent more time freezing during the context test, when compared to NS and HVS groups. However, in groups that received post-stress glucose, both MVS and HVS groups exhibited freezing levels higher than the NS group. Regardless of fluid condition, both the HVS and MVS groups showed significantly less weight gain when compared to the NS group. Error bars denote mean ± SEM. * p < .05, ** p < .01, *** p < .001, **** p < .0001.

effects of Stress, F(2, 40) = 12.34, $p$ < .0001, and Fluid, F (1, 40) = 4.945, $p$ = .0319. Tukey post-hoc comparisons on stress indicated a relationship among groups, such that NS > MVS = HVS. A one-way ANOVA showed no statistically significant effect of group on shock reactivity, F(2, 21) = 1.622, p = .221, or baseline freezing (F < 1).

### 3.4 2-deoxy-D-glucose-induced glucoprivation inhibits the formation of contextual SEFL behavior in rats exposed to MVS

The previous experiment showed that consumption of a glucose solution is enough to produce contextual SEFL in rats exposed to HVS, which otherwise do not exhibit the phenotype. Here we test the opposite: is artificial glucose deprivation sufficient to inhibit the expression of SEFL in MVS animals? Previous research has shown that glucoprivation induced by 2-deoxy-D-glucose (2DG) in the absence of stress was sufficient to induce several of the behavioral phenotypes typically observed following HVS [28]. We test the hypothesis that injection of 2DG will suppress the expression of the SEFL phenotype in rats exposed to MVS stress.

Fig 4 shows percent freezing during the contextual fear test (A) and weight change (B) among groups one week following stress pretreatment. Rats that received vehicle and MVS showed higher freezing levels compared to the NS group during the context test 24 hours after 1-shock exposure (as previously seen). However, the group exposed to MVS and given an injection of 2-DG prior to fear conditioning exhibited decreased fear expression compared to their vehicle-injected counterparts (A). A one-way ANOVA on freezing during the context test yielded a significant main effect of Group, F(2, 37) = 10.740, p = .0002. Tukey post-hoc comparisons on group means indicated a relationship among groups, such that: NS-V = MVS-D <

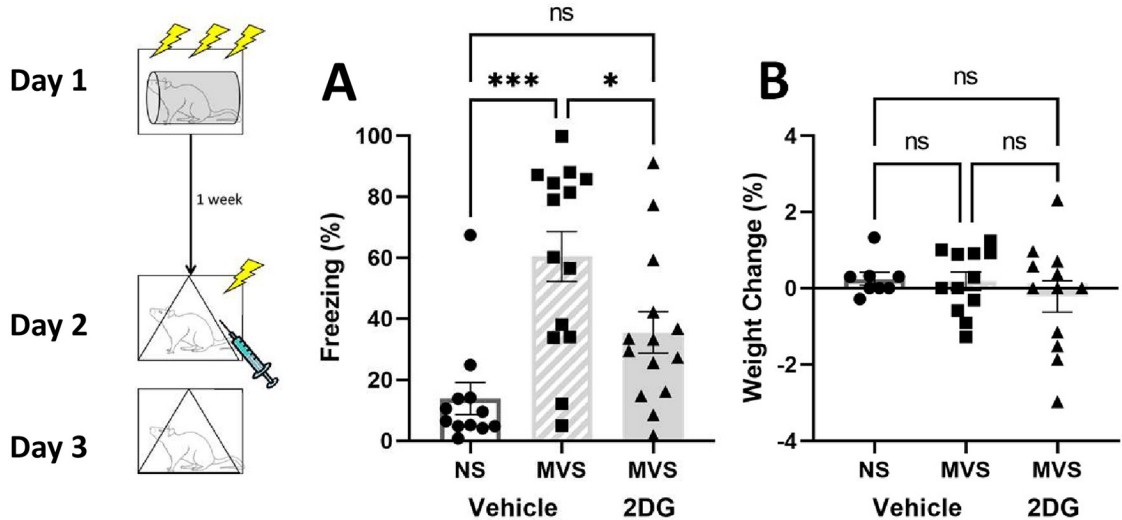

**Fig 4. Impacts of 2-deoxy-d-glucose injection on SEFL and weight maintenance in rats exposed to MVS.** Depicted: Percent freezing during context conditioning test (panel A) and weight change (panel B). Rats were exposed to 0 (NS), or 15 (MVS) tail shocks one week prior to fear conditioning testing. All groups received intraperitoneal injection of 2DG or vehicle prior to fear conditioning. Testing consisted of exposure to a single, 1 mA shock in a novel context. Rats were then returned to this same context 24-hours later. Rats were weighed prior to stress exposure and prior to fear conditioning testing. When 2DG was given prior to fear conditioning, the MVS-vehicle group spent more time freezing during the context group when compared to the NS and MVS-2DG groups. No weight maintenance effects of 2DG were observed at the time of testing. Error bars denote mean ± SEM. * p < .05, *** p < .001.

MVS-V. No significant differences in weight change were observed among groups (B), $F_{(2, 29)}$ = 0.6203, p = .545. A one-way ANOVA showed no statistically significant effect of group on shock reactivity or baseline freezing to the 1-shock context (F < 1).

### 3.5 Moderate and high-volume stressors result in dissociable neurobiological effects

We have shown that stress volume impacts the subsequent behavioral phenotype in a dissociable manner. Here, we tested the hypothesis that stress also produces dissociable neurobiological effects. Our lab has previously shown that MVS stress increases GluA1 expression in the BLA [35]. Since we have shown that HVS inhibits hippocampal-dependent (context) SEFL, we hypothesized that HVS will produce a reduction in N-methyl-D-aspartate (NMDA) receptor concentrations in the hippocampus. We also hypothesized that HVS will induce a similar increase of GluA1 in the BLA, since HVS enhanced hippocampal-independent auditory SEFL.

Fig 5 shows AMPA and NMDA receptor subunit protein quantification in the BLA (E-H) and the DH (I-L) one week after stress treatment. As previously seen, rats exposed to HVS exhibited greater weight loss seven days after stress exposure (B). A one-way ANOVA on Weight Change (%) yielded a significant main effect of Group, $F_{(3,20)}$ = 4.413, p = .0155. Tukey post-hoc comparisons (α = .05) on Weight Change indicated the following ordered relationship among means: HCC = NS = MVS > HVS.

Rats exposed to MVS or HVS exhibited greater levels of GluA1 in the BLA compared to HCC (E); rats exposed to HVS, but not MVS, also exhibited higher levels of GluA2 in the BLA compared to HCC (F). One-way ANOVAs on BLA protein analysis yielded significant main effects of Group on GluA1/GAPDH, $F_{(3, 20)}$ = 15.93, $p < .0001$, and GluA2/GAPDH, $F_{(3, 20)}$ = 3.214, $p = .0449$. Tukey post-hoc comparisons (α = .05) on GluA1/GAPDH indicated the following ordered relationship among group means: HCC = NS < MVS < HVS. Tukey post-hoc

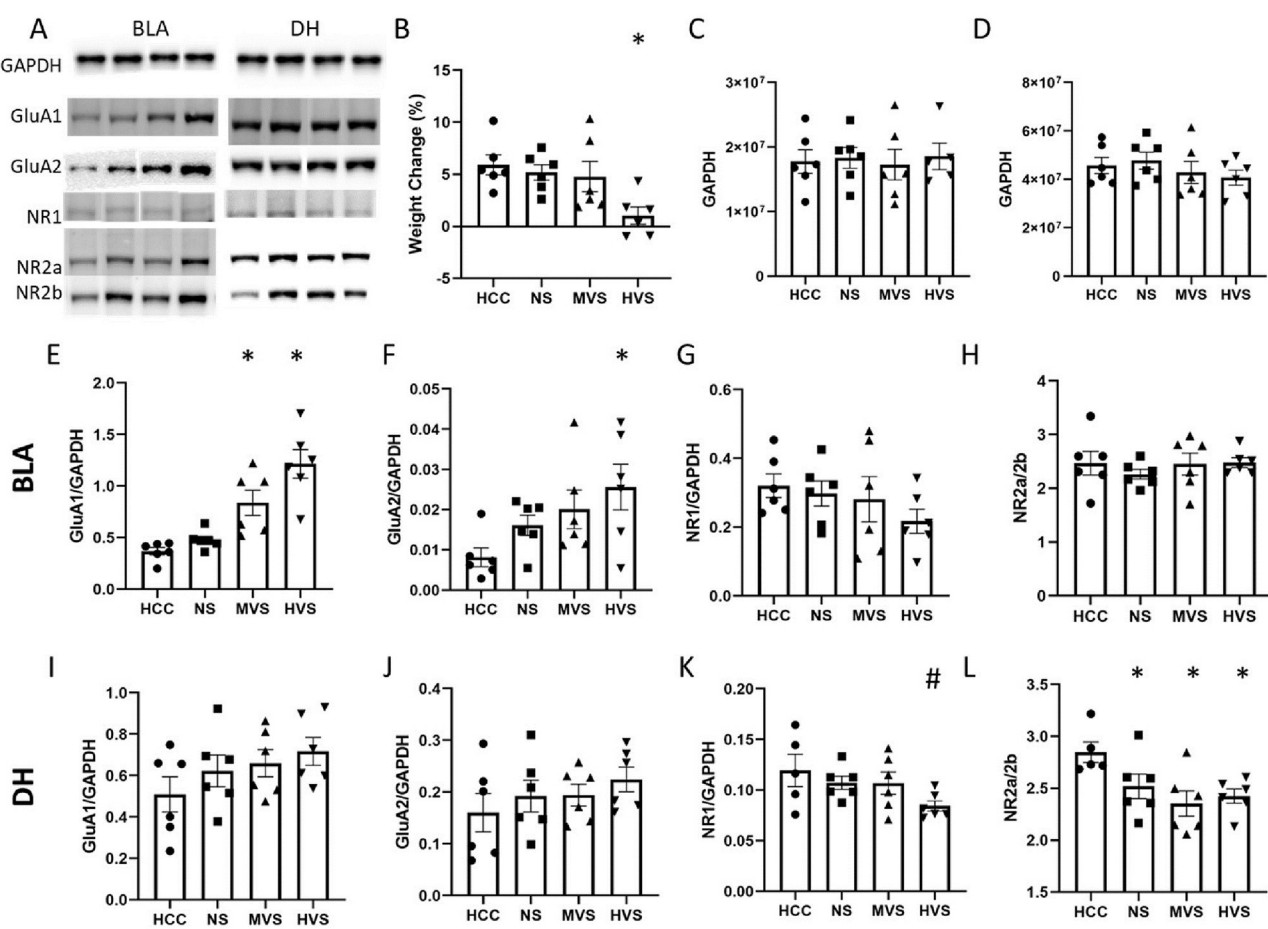

**Fig 5. Neurobiological effects of stress volume.** Depicted: BLA (panels E-H) and DH (panels I-L) concentrations of GluA1 (panels E & I), GluA2 (panels F & J), NR1 (panels G & K), and NR2a/2b (panels H & L) as determined by western blot analysis. GluA1, GluA2, and NR1 are depicted as a ratio over GAPDH concentrations (panels C & D). Rats were exposed to 0 (NS), 15 (MVS), or 100 (HVS) tailshocks, or remained in their home cage (HCC) never exposed to context or a stressor, one week prior to sacrifice for tissue analysis. Rats were weighed prior to stress exposure and prior to sacrifice. MVS and HVS groups exhibited higher BLA concentrations of GluA1 compared to HCC and NS groups. The HVS exhibited higher concentrations of GluA2 compared to the HCC group. The HVS group had a lower concentration of DH NR1 when compared to NS. All groups exhibited a lower NR2a/2b ration in the DH when compared to the HCC group. Error bars denote mean ± SEM. * p < .05 (compared to HCC), # p < .05 (compared to NS).

comparisons ($\alpha$ = .05) on GluA2/GAPDH indicated the following ordered relationship among group means: HCC < HVS. No group differences were observed in BLA NR1 or NR2a/2b ratios (G & H).

No group differences in DH GluA1 or GluA2 were observed (I & J). However, rats exposed to HVS exhibited decreased concentrations of NR1 in the DH compared to NS controls (K). Furthermore, all stressed groups exhibited a decreased NR2a:2b ratio in the DH compared to HCC (L). One-way ANOVAs on DH protein analysis yielded a significant main effect of Group on NR2a/2b, $F(3, 20)$ = 3.980, $p$ = .0234. Tukey post-hoc comparisons ($\alpha$ = .05) on NR2a/2b indicated the following ordered relationship among group means: HCC > NS = MVS = HVS. Due to high variability in the HCC, any group effect on DH NR1 was not statistically significant. However, if the HCC group is removed from the analysis, a one-way ANOVA on DH protein analysis yields a significant main effect of Group on NR1, $F(2, 14)$ = 4.651, $p$ = .0283. Tukey post-hoc comparisons on NR1 indicated the following ordered relationship

among group means: NS = MVS > HVS. No significant main effects of Group were found during protein analysis of the VH.

## 4. Discussion

The experiments described above provide evidence that the volume of a stressor is a key factor in determining the behavioral and neurobiological consequences of stress and, this cannot simply be summarized as more stress leads to greater deleterious effects. We found evidence that supports the notion that HVS may model stress-induced conditions that have a depression component or comorbidity, while MVS may better model anxiety-only disorders. Furthermore, we found evidence that suggests that glucose exerts its behavioral effects exclusively in high volume-stressed rats. The effects of glucose appear to not only eliminate HVS-induced phenotypy [24–26], but in the case of SEFL, facilitate it in HVS-stressed animals. This is further exemplified by our finding that when rats exposed to MVS were glucoprivated during single-trial conditioning, the SEFL effect was eliminated. Finally, we provide evidence that stressors of different volumes produce dissociable changes in AMPA and NMDA receptor density in the BLA and dorsal hippocampus.

There are a number of potential mechanisms through which stress volume exerts its effects on subsequent fear learning. One possible explanation is that HVS is producing a general deficit in contextual fear learning that masks the sensitization effect of MVS. Contextual learning critically depends on hippocampal processing [42, 43, 49] and the hippocampus is profoundly affected by stress [50, 51]. An increase in circulating glucocorticoids during stress impairs glucose uptake transport into the hippocampus and severely impairs contextual processing [52–56]. The HVS procedure used in our experiments produces deficits in contextual discrimination [40] and long-term effects on hippocampal spine density [57], neurogenesis [38], synaptic plasticity, and long-term potentiation [37, 58]. Deficits in contextual learning are reversed by increasing hippocampal glucose concentrations by several means [59–61]. Therefore, while HVS may still induce the non-associative fear sensitization process within the BLA that occurs in MVS, the behavioral expression of this process may be nullified by an overall decrease in contextual fear learning (see Fig 6). This is, in part, supported by our finding that HVS *did* enhance fear conditioning to a tone. In addition, evidence suggests that cued fear conditioning is hippocampal-independent [44, 62]. While there are some conditions where the hippocampus plays a role in fear elicited by auditory cues, that effect is typically seen after CS termination [45] and our measure was confined to the 30 sec tone presentation. Our finding that cued, but not contextual, fear conditioning is enhanced by HVS therefore may suggest that hippocampal functioning may be impaired by exposure to an HVS. This hypothesis is further supported by our finding that HVS, but not MVS, decreases NR1 expression in the DH. NR1 is the obligatory NMDA receptor subunit and provides a reasonable estimate for NMDA receptor concentration [63]. Hippocampal NMDA receptor activity is essential for the acquisition of contextual fear [49, 62, 64]. Therefore, stress-enhanced contextual fear learning may be inhibited in HVS by decreasing the hippocampus' ability to form new contextual memories. Follow-up studies investigating the differential impacts of MVS and HVS on peripheral and hippocampal energy homeostasis will be an important step in assessing this proposed mechanism. We have previously found that HVS uniquely challenges peripheral energy homeostasis, but these effects have not been compared to the effects following MVS exposure [24].

These results present but a few examples of how the behavioral and biological outcomes of stress can be counter-intuitive. These studies explore the outer extremes of stress volume, and follow-up exploring intermediary values is clearly necessary. Furthermore, while we controlled for several factors, several procedural differences did remain. For example, while the MVS and

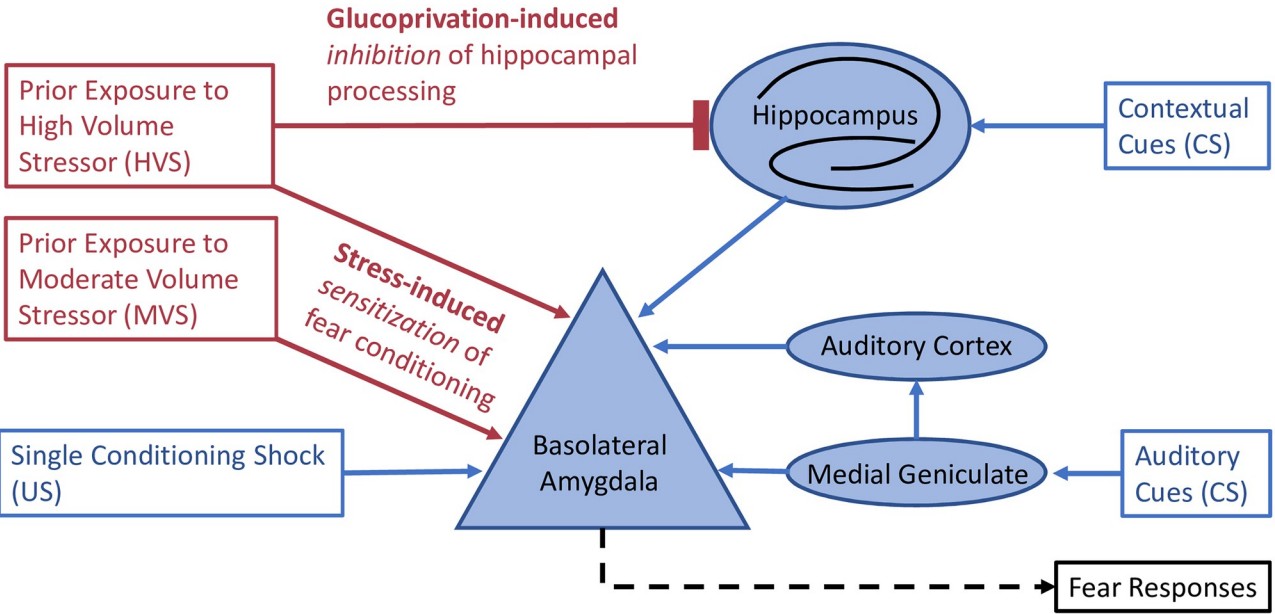

**Fig 6. Schematic of hypothesized mechanism.**

HVS exposures occur over a relatively similar timeframe (90 and 114 minutes, respectively), this necessitates that the intervals between shocks are vastly different (six minutes and one minute, respectively). The length of individual shocks is also different between procedures. These aspects of shock undoubtedly impact the subsequent behavior of the animal (in fact, see [65–67]) and should also be parametrically studied. It is important to note that this study reports effects on male rats exclusively. This non-trivial limitation was necessary, as the LH model has historically used males due to the intriguing, but confounding, resiliency in female rats [68]. Therefore, these findings cannot be extended to female rats, and potential sex differences must be explored further. Finally, this is the first study to demonstrate tailshock-to-footshock SEFL using the 15-shock design. Therefore, this study provides further evidence that SEFL is not simply a generalization of the footshock event carrying over to the single-trial conditioning session.

The effects of stress exposure are variable in severity and quality [69, 70]. Treatment efficacy for patients suffering from stress-induced psychiatric disease continues to remain similarly variable, with only a small percent of the population seeing a persistent quelling of symptoms [71, 72]. One hypothesis is that the observed variability in treatment effectiveness may correlate with variability in stress exposure; the quality, severity, and chronicity of the experienced trauma may have a direct impact on symptoms expressed and the probability of a positive treatment outcome. While parametric study of trauma exposure is impossible in a clinical population, animal models provide us with the necessary tools to interrogate the effects of stress dimensions through direct, controlled comparison. Surprisingly, the experiments described in this manuscript are some of the first to directly study the effects of stress volume using appropriate controls. Therefore, despite an enormous literature devoted to the effects of stress, we are one of the first to provide evidence that the dimensions of the stressor used can directly impact the subsequent behavioral and biological profiles. The eventual goal of this (and future) effortful, parametric work on the effects of various stress dimensions is to provide insight on how information regarding the type of trauma experienced can help inform predictions on disease development and treatment.

## Author Contributions

**Conceptualization:** Michael A. Conoscenti, Michael S. Fanselow.

**Data curation:** Michael A. Conoscenti.

**Formal analysis:** Michael A. Conoscenti.

**Funding acquisition:** Michael S. Fanselow.

**Investigation:** Nancy J. Smith.

**Methodology:** Michael A. Conoscenti, Michael S. Fanselow.

**Resources:** Michael S. Fanselow.

**Supervision:** Michael S. Fanselow.

**Visualization:** Michael A. Conoscenti.

**Writing – original draft:** Michael A. Conoscenti.

**Writing – review & editing:** Michael A. Conoscenti, Nancy J. Smith, Michael S. Fanselow.

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
