## [Decision Letter · Decision Letter 0]

20 May 2022

PONE-D-22-09431Dissociable consequences of moderate and high volume stress are mediated by the differential energetic demands of stress.PLOS ONE

Dear Dr. Conoscenti,

Thank you for submitting your manuscript to PLOS ONE. After careful consideration, we feel that it has merit but does not fully meet PLOS ONE’s publication criteria as it currently stands. Therefore, we invite you to submit a revised version of the manuscript that addresses the points raised during the review process. The reviewers both were both positive about the work but raised several issues that should be addressed in a revision.

We look forward to receiving your revised manuscript.

Kind regards,

Sayamwong E. Hammack, Ph.D

Academic Editor

PLOS ONE

Journal Requirements:

"This research was supported by NIH Grant R01-MH115678 and funds from the Staglin Center for Brain & Behavioral Health."

"This research was supported by NIH Grant R01-MH115678 (M.S.F.) and funds from the Staglin Center for Brain & Behavioral Health (M.S.F.). The funders had no role in study design, data collection and analysis, decision to publish, or preparation of the manuscript."

Reviewers' comments:

Reviewer's Responses to Questions

**Comments to the Author**

1. Is the manuscript technically sound, and do the data support the conclusions?

Reviewer #1: Yes

Reviewer #2: Yes

2. Has the statistical analysis been performed appropriately and rigorously? 

Reviewer #1: Yes

Reviewer #2: Yes

3. Have the authors made all data underlying the findings in their manuscript fully available?

Reviewer #1: Yes

Reviewer #2: Yes

4. Is the manuscript presented in an intelligible fashion and written in standard English?

Reviewer #1: Yes

Reviewer #2: Yes

5. Review Comments to the Author

Reviewer #1: The manuscript reports a series of experiments examining how “shock volume” impacts glucose-mediated fear learning. In the manuscript, the authors contrast three groups: High-volume shock (HVS; 100 shocks), moderate-volume shock (MVS; 15 shocks) and no-shock controls (NS). In the first experiment there was a double dissociation between shock volume and contextual fear and weight gain. MVS increased contextual fear and had no impact on weight gain, whereas HVS had no impact on fear but decreased weight gain. Both HVS and MVS increased expression of auditory fear conditioning. Next, rats that consumed a glucose solution also showed enhanced contextual fear relative to NS, and glucosprivated rats did not show enhanced contextual fear following MVS. Thus, the authors show bi-directional control of this the impact of MVS on contextual fear. Finally, the authors report dissociable effects of glutamate receptor expression in the BLA and DH.

Overall, this is an excellent series of experiments. As mentioned earlier, it is particularly thoughtful to see effects manipulated in both directions. I have one major comment, and several more minor comments.

Major: My main concern is related to the cued fear conditioning experiment. The was a nice addition; it allowed the authors to examine if the MVS vs. HVS dissociation was general to fear expression or specific to the expression of contextual fear learning. In this experiment, while MVS increased contextual fear, and HVS did not (relative to NS), expression of cued fear was increased for both MVS and HVS.

The concern I have is that the authors describe the tone-shock association as not requiring the hippocampus. While it is certainly true that there are many reports that delay fear conditioning is not impacted by pre- or post-training damage to the hippocampus, there is additional research that suggests this may depend on the “strength” of the association. Indeed, Quinn et al. (2008) demonstrate that when the tone-shock association is weak, then expression of auditory fear conditioning does depend on the hippocampus. Given that in the current paper there was only one pairing of the tone and shock, it seems reasonable that this would be considered a weak association.

I’m sure the authors are aware of this, and likely have some thoughts. I don’t think this needs to be addressed with additional experiments. But I wonder if the authors could address this issue. Although it would be a strong argument if indeed this tone-shock association were hippocampal independent, I do think it is still a compelling dissociation.

Minor:

1) Pg. 7. The experiments were run in only male rats. However, the authors provide suitable logic for this in the general discussion.

2) All the figures were “fuzzy”. This might have been an issue with my own printer, or maybe with the pre-processing by the journal. I only mention it just so the authors are aware in case it is related to the resolution they were saved as.

3) Pg. 8 and 9. In general the reader could use some more details regarding the procedures of the experiment. As just one example, the authors note that rats were pre-exposed to a novel context, but they don’t mention duration of exposure.

4) Pg. 11. The authors report an ANOVA on post-shock freezing. But I think they are referring to the retrieval test?

5) Pg. 11. The results for figure 1 might be re-ordered. For the reader, it might be more logical to discuss panel A then B then C then D. I thought this was also true for the way the data is described for Figure 2.

6) I like the additional analysis of shock reactivity and baseline freezing. I think the authors can decide – but would it be worth it to include in the figures?

7) For brevity, when an F value is less than 1, I think the authors could report (F < 1) as opposed to the full value (e.g, F = 0.006837).

8) Figure 2 panel C. Trials on the X-axis. This this 1-minute time bins?

Reviewer #2: The present manuscript (PONE-D-22-09431) examines the impact of different doses of tailshock (15 vs 100 shock procedures) on subsequent fear conditioning and glutamate receptor protein expression. The authors state that the 15-shock procedure is analogous to the stress-enhanced fear learning model and the 100-shock procedure is analogous to the learned helplessness paradigm. The authors present a reasonable hypothesis that the two procedures generate different molecular and behavioral sequelae due to differences in metabolic demand.

The hypothesis would be strengthened if the authors provide a comparison of post-stress metabolic measures between the two stress types (moderate vs high-volume stress). Giving 2-deoxy-D-glucose is not the same as directly measuring energy homeostasis. Furthermore, do the authors have any evidence that the glucose manipulations that alter the behavioral phenotype also alter their hippocampal receptor subunit findings?

One issue with framing the present results in the context of stress-enhanced fear learning (SEFL) versus learned helplessness (LH) is that the authors don’t accurately define what learned helplessness actually is. LH refers to stressor outcomes that depend on the uncontrollability of the stressor. That is, to qualify as a LH effect an outcome must follow exposure to inescapable (IS), but not physically identical escapable (ES), shocks or other aversive events. To ignore the controllability issue is to make LH effects synonymous with generic stress effects, which they are not. There are plenty of outcomes (e.g., neurochemical, behavioral) of tail shock that are not sensitive to the dimension of controllability. Thus, there can be many reasons for poor shuttlebox escape responding, only one of which is learned helplessness.

Related to above, the authors imply that the gold standard for LH experiments is to deliver 100 x 1.0 mA shocks for an average length of 8 seconds (800 seconds of total shock). The majority of published LH studies (those that include inescapable and escapable groups) use considerably less shock volume. Mean wheel-turn escape times across 100 shocks are around 3-5 seconds (300-500 seconds of total shock). If the authors frame their high-volume stress treatment as a typical learned helplessness design, then they should provide references demonstrating that their shock parameters produce effects that are selective to the uncontrollability of the stressor.

The claim that these results represent “the first study to demonstrate tailstock-to-footshock” stress-enhanced fear learning ignores a number of published papers showing that tail shock enhances fear learning (both cued and contextual). As an example, uncontrollable tail shock leads to enhanced footshock-elicited freezing in a shuttle box 24 hr later (plenty of studies from the Maier laboratory), a conditional response attributable to contextual cues of the shuttle box apparatus (Fanselow, 1980). In fact, the majority of studies show that uncontrollable tail shock increases, rather than interferes with, contextual fear.

The abstract states that weight gain was impacted only in high-volume stress animals. Was this a consistent finding throughout the studies? It appears that weight gain was depressed in both high and moderate groups compared to No Stress in Figure 3. And in Figure 4B the % weight change for the MVS-Veh group is similar to that of the HVS group in Figure 1D.

6. PLOS authors have the option to publish the peer review history of their article (what does this mean?). If published, this will include your full peer review and any attached files.

Reviewer #1: No

Reviewer #2: No

---

## [Author Response · Author response to Decision Letter 0]

10 Jun 2022

"This research was supported by NIH Grant R01-MH115678 and funds from the Staglin Center for Brain & Behavioral Health."

"This research was supported by NIH Grant R01-MH115678 (M.S.F.) and funds from the Staglin Center for Brain & Behavioral Health (M.S.F.). The funders had no role in study design, data collection and analysis, decision to publish, or preparation of the manuscript."

The above funding statement is correct. As such there are no changes necessary. For financial disclosures, however, we would like to add “Financial Disclosures-MSF is a board member of Neurovation Inc. The authors declare that the research was conducted in the absence of any commercial or financial relationships that could be construed as a potential conflict of interest.” The acknowledgment and financial disclosure sections have been removed, per request, to avoid redundancy.

It appears we failed to note the github link somewhere within the submission, our apologies. It is available at https://github.com/mconoscenti/shock_volume. This has also been noted in the revised cover letter.

The raw Western Blot data has been uploaded to Github as well, https://github.com/mconoscenti/shock_volume/tree/mconoscenti-westernblot-raw. 

Reviewers' comments:

Reviewer's Responses to Questions

Comments to the Author

Reviewer #1: The manuscript reports a series of experiments examining how “shock volume” impacts glucose-mediated fear learning. In the manuscript, the authors contrast three groups: High-volume shock (HVS; 100 shocks), moderate-volume shock (MVS; 15 shocks) and no-shock controls (NS). In the first experiment there was a double dissociation between shock volume and contextual fear and weight gain. MVS increased contextual fear and had no impact on weight gain, whereas HVS had no impact on fear but decreased weight gain. Both HVS and MVS increased expression of auditory fear conditioning. Next, rats that consumed a glucose solution also showed enhanced contextual fear relative to NS, and glucosprivated rats did not show enhanced contextual fear following MVS. Thus, the authors show bi-directional control of this the impact of MVS on contextual fear. Finally, the authors report dissociable effects of glutamate receptor expression in the BLA and DH.

Overall, this is an excellent series of experiments. As mentioned earlier, it is particularly thoughtful to see effects manipulated in both directions. I have one major comment, and several more minor comments.

Major: My main concern is related to the cued fear conditioning experiment. The was a nice addition; it allowed the authors to examine if the MVS vs. HVS dissociation was general to fear expression or specific to the expression of contextual fear learning. In this experiment, while MVS increased contextual fear, and HVS did not (relative to NS), expression of cued fear was increased for both MVS and HVS.

The concern I have is that the authors describe the tone-shock association as not requiring the hippocampus. While it is certainly true that there are many reports that delay fear conditioning is not impacted by pre- or post-training damage to the hippocampus, there is additional research that suggests this may depend on the “strength” of the association. Indeed, Quinn et al. (2008) demonstrate that when the tone-shock association is weak, then expression of auditory fear conditioning does depend on the hippocampus. Given that in the current paper there was only one pairing of the tone and shock, it seems reasonable that this would be considered a weak association.

I’m sure the authors are aware of this, and likely have some thoughts. I don’t think this needs to be addressed with additional experiments. But I wonder if the authors could address this issue. Although it would be a strong argument if indeed this tone-shock association were hippocampal independent, I do think it is still a compelling dissociation.

This really is an interesting worth acknowledgement in the paper. It is true that out single-trial conditioning would reasonably fall within a procedure that produces a weak association. Indeed, that’s why the procedure is chosen! A weak association allows for very clear SEFL effects. However, in a sufficiently stressed animal, a single trial no longer produces a weak association. In the 100 shock group, however, since they are not exhibiting high levels of contextual freezing, it becomes a much larger puzzle to determine if our single-trial conditioning should be considered a procedure that nourishes a weak or strong association. 

In an effort to remain succinct within the manuscript, we have added a note regarding the Quinn et al (2008) series of experiments, but have not gone into the weeds regarding this issue. Note that in the Quinn study hippocampal lesion effects were typically seen after, not during CS presentation, the data we presented were all during CS presentation. We have also softened claims suggesting that this experiment comprehensively tests this hypothesis.

Minor:

1) Pg. 7. The experiments were run in only male rats. However, the authors provide suitable logic for this in the general discussion.

Thank you for noting this important point.

2) All the figures were “fuzzy”. This might have been an issue with my own printer, or maybe with the pre-processing by the journal. I only mention it just so the authors are aware in case it is related to the resolution they were saved as.

Thank you for pointing this out. The files were uploaded at high resolution, but we will be sure that the figures look clear during the preprint process.

3) Pg. 8 and 9. In general the reader could use some more details regarding the procedures of the experiment. As just one example, the authors note that rats were pre-exposed to a novel context, but they don’t mention duration of exposure.

Information regarding preexposure has been added. Other clarifying language regarding procedures and reorganization of the method section has been performed to better spell out all procedures performed.

4) Pg. 11. The authors report an ANOVA on post-shock freezing. But I think they are referring to the retrieval test?

Yes, that is correct. Wording has been changed to accurately reflect this.

5) Pg. 11. The results for figure 1 might be re-ordered. For the reader, it might be more logical to discuss panel A then B then C then D. I thought this was also true for the way the data is described for Figure 2.

This point is appreciated. We have reordered the results of B & D to read in the appropriate order. We have kept the discussion of A & C together, however, to avoid seeming unnecessarily redundant by discussing the results of 1-day and 1-week separately where they are nearly identical.

6) I like the additional analysis of shock reactivity and baseline freezing. I think the authors can decide – but would it be worth it to include in the figures?

These data were included in initial internal drafts. However, because much of the baseline freezing data was zero, it felt unnecessary to provide graphs that were essentially empty.

7) For brevity, when an F value is less than 1, I think the authors could report (F < 1) as opposed to the full value (e.g, F = 0.006837).

Where applicable, the full reporting has been changes to simply “F < 1”.

8) Figure 2 panel C. Trials on the X-axis. This this 1-minute time bins?

The x-axis denotes the three preexposure trials. Hopefully, with the additional information regarding our preexposure procedure, this becomes clearer to the reader. We have also included further description in the figure legend.

Reviewer #2: The present manuscript (PONE-D-22-09431) examines the impact of different doses of tailshock (15 vs 100 shock procedures) on subsequent fear conditioning and glutamate receptor protein expression. The authors state that the 15-shock procedure is analogous to the stress-enhanced fear learning model and the 100-shock procedure is analogous to the learned helplessness paradigm. The authors present a reasonable hypothesis that the two procedures generate different molecular and behavioral sequelae due to differences in metabolic demand.

The hypothesis would be strengthened if the authors provide a comparison of post-stress metabolic measures between the two stress types (moderate vs high-volume stress). Giving 2-deoxy-D-glucose is not the same as directly measuring energy homeostasis. Furthermore, do the authors have any evidence that the glucose manipulations that alter the behavioral phenotype also alter their hippocampal receptor subunit findings?

Post-stress metabolic measures would indeed help illuminate the potential mechanism of these effects. Indeed, we have previously reported that HVS greatly challenges metabolic homeostasis (Conoscenti, et al. 2019). However, these types of studies are extensive and we felt were beyond the purview of the current study. However, getting into the physiological mechanisms of the reported effects is an excellent future direction that has been added to the discussion section. We did indeed attempt to examine the effects of glucose on the receptor subunit findings in the same animals that underwent behavioral testing. We have found, however, that subsequent testing (ie. fear conditioning) shrouds the stress effects, and the data were therefore inconclusive.

One issue with framing the present results in the context of stress-enhanced fear learning (SEFL) versus learned helplessness (LH) is that the authors don’t accurately define what learned helplessness actually is. LH refers to stressor outcomes that depend on the uncontrollability of the stressor. That is, to qualify as a LH effect an outcome must follow exposure to inescapable (IS), but not physically identical escapable (ES), shocks or other aversive events. To ignore the controllability issue is to make LH effects synonymous with generic stress effects, which they are not. There are plenty of outcomes (e.g., neurochemical, behavioral) of tail shock that are not sensitive to the dimension of controllability. Thus, there can be many reasons for poor shuttlebox escape responding, only one of which is learned helplessness.

We initially reference learned helplessness not as a means to tie the current results with the phenomena, but merely to provide a justification for the stress procedure used. We agree that it is impossible to determine LH without the escapable group. However, we also know that the effect is somewhat unique in that it requires a significant exposure to (uncontrollable and unpredictable) stress. Whether that be 300-800 seconds of total shock, this is still far above the level of shock required for many other behavioral effects (such as SEFL). It therefore provides us with an excellent procedural comparison to more moderate stressors that do not produce these unique shuttle-escape deficit effects. To be clear, here we are looking at the effects that two types of stress have on behavior and are not exploring how control can reduce the effects of stress. Wording stressing this effect has been added to the introduction.

Additionally, we had previously attempted to separate LH from HVS as quickly as possible, while still acknowledging it as a source of inspiration. To extend this end, we have further refined this section of the introduction to further clarify that we are making no statements about LH per se, but rather interested in the 100 shock procedure, originally tied to LH, that produces an array of unique effects.

Related to above, the authors imply that the gold standard for LH experiments is to deliver 100 x 1.0 mA shocks for an average length of 8 seconds (800 seconds of total shock). The majority of published LH studies (those that include inescapable and escapable groups) use considerably less shock volume. Mean wheel-turn escape times across 100 shocks are around 3-5 seconds (300-500 seconds of total shock). If the authors frame their high-volume stress treatment as a typical learned helplessness design, then they should provide references demonstrating that their shock parameters produce effects that are selective to the uncontrollability of the stressor.

This point is noted and appreciated. The description of canonical shock duration has been adjusted to a range to reflect this point.

The claim that these results represent “the first study to demonstrate tailstock-to-footshock” stress-enhanced fear learning ignores a number of published papers showing that tail shock enhances fear learning (both cued and contextual). As an example, uncontrollable tail shock leads to enhanced footshock-elicited freezing in a shuttle box 24 hr later (plenty of studies from the Maier laboratory), a conditional response attributable to contextual cues of the shuttle box apparatus (Fanselow, 1980). In fact, the majority of studies show that uncontrollable tail shock increases, rather than interferes with, contextual fear.

The reviewer is absolutely right on this point. We were really intending this statement to be specific to the canonical SEFL design (15 shock stressor followed by 1-trial conditioning) in order to address previous critiques of the model. This has been reworded to clarify this claim.

The abstract states that weight gain was impacted only in high-volume stress animals. Was this a consistent finding throughout the studies? It appears that weight gain was depressed in both high and moderate groups compared to No Stress in Figure 3. And in Figure 4B the % weight change for the MVS-Veh group is similar to that of the HVS group in Figure 1D.

This finding was consistent across all studies (ie. see figure 5), except for the study where the glucose manipulation was present (though the general trend here is consistent). We did find it interesting in the case of the glucose study, and our best guess was that the preexposure to glucose provides its own (albeit opposite) challenge to energy homeostasis which may be somewhat hormetic to future challenges. This may be enough to obfuscate any stress effects on weight. This experiment also required several cohorts due to its size. Therefore it’s also possible that the variability between cohorts was enough to mitigate the statistical power between stress groups. As for Figure 4B, the control group suggests that this cohort may have generally been slower at gaining weight.

---

## [Decision Letter · Decision Letter 1]

5 Jul 2022

PONE-D-22-09431R1Dissociable consequences of moderate and high volume stress are mediated by the differential energetic demands of stress.PLOS ONE

Dear Dr. Conoscenti,

Thank you for submitting your manuscript to PLOS ONE. After careful consideration, we feel that it has merit but does not fully meet PLOS ONE’s publication criteria as it currently stands. Therefore, we invite you to submit a revised version of the manuscript that addresses the points raised during the review process.

One of the reviewers still had some minor comments that should be addressed.==============================

We look forward to receiving your revised manuscript.

Kind regards,

Sayamwong E. Hammack, Ph.D

Academic Editor

PLOS ONE

Journal Requirements:

Reviewers' comments:

Reviewer's Responses to Questions

**Comments to the Author**

1. If the authors have adequately addressed your comments raised in a previous round of review and you feel that this manuscript is now acceptable for publication, you may indicate that here to bypass the “Comments to the Author” section, enter your conflict of interest statement in the “Confidential to Editor” section, and submit your "Accept" recommendation.

Reviewer #1: All comments have been addressed

Reviewer #2: (No Response)

2. Is the manuscript technically sound, and do the data support the conclusions?

Reviewer #1: Yes

Reviewer #2: Yes

3. Has the statistical analysis been performed appropriately and rigorously? 

Reviewer #1: Yes

Reviewer #2: Yes

4. Have the authors made all data underlying the findings in their manuscript fully available?

Reviewer #1: Yes

Reviewer #2: Yes

5. Is the manuscript presented in an intelligible fashion and written in standard English?

Reviewer #1: Yes

Reviewer #2: Yes

6. Review Comments to the Author

Reviewer #1: (No Response)

Reviewer #2: The authors still use the term “learned helplessness” in a way that is synonymous with “stress”. The term is not appropriate for simply any behavioral or physiological consequence of uncontrollable tailshock. Rather, it is appropriate for behavioral changes that are specifically mediated by the inescapable/uncontrollable nature of the shock, and not the shock itself. The distinction is important since the motivation of the present experiments (Intro), the hypothesis (line 171), and experimental design (line 208) are framed within the context of LH.

Statements such as “Rats exposed to LH experience a total of about 800 seconds of 1 mA shock (100 shocks at an average of 8 seconds each)” and “HVS and MVS parameters were chosen to mimic previously published work on LH…” are not supported by any reference that shows that these parameters produce learned helplessness. Certainly they produce stress effects, but not necessarily LH.

7. PLOS authors have the option to publish the peer review history of their article (what does this mean?). If published, this will include your full peer review and any attached files.

Reviewer #1: No

Reviewer #2: No

---

## [Author Response · Author response to Decision Letter 1]

19 Jul 2022

The authors still use the term “learned helplessness” in a way that is synonymous with “stress”. The term is not appropriate for simply any behavioral or physiological consequence of uncontrollable tailshock. Rather, it is appropriate for behavioral changes that are specifically mediated by the inescapable/uncontrollable nature of the shock, and not the shock itself. The distinction is important since the motivation of the present experiments (Intro), the hypothesis (line 171), and experimental design (line 208) are framed within the context of LH.

Statements such as “Rats exposed to LH experience a total of about 800 seconds of 1 mA shock (100 shocks at an average of 8 seconds each)” and “HVS and MVS parameters were chosen to mimic previously published work on LH…” are not supported by any reference that shows that these parameters produce learned helplessness. Certainly they produce stress effects, but not necessarily LH.

- We have further clarified this point. As stated before, it is not our intention to draw conclusions regarding learned helplessness, or draw a misleading premise. Specifically, we have added language in the into to reinforce the point that the HVS stress is simply a stressor that is parametrically similar to the LH-inducing stressor. We have also added clarifying language to the hypothesis line and experimental design to avoid any further confusion on this point.

---

## [Editor Report · Decision Letter 2]

16 Aug 2022

Dissociable consequences of moderate and high volume stress are mediated by the differential energetic demands of stress.

PONE-D-22-09431R2

Dear Dr.Conoscenti,

We’re pleased to inform you that your manuscript has been judged scientifically suitable for publication and will be formally accepted for publication once it meets all outstanding technical requirements.

Kind regards,

Sayamwong E. Hammack, Ph.D

Academic Editor

PLOS ONE
---

## [Editor Report · Acceptance letter]

23 Aug 2022

PONE-D-22-09431R2 

Dissociable consequences of moderate and high volume stress are mediated by the differential energetic demands of stress. 

Dear Dr. Conoscenti:

I'm pleased to inform you that your manuscript has been deemed suitable for publication in PLOS ONE. Congratulations! Your manuscript is now with our production department. 

Kind regards, 

on behalf of

Dr. Sayamwong E. Hammack 

Academic Editor

PLOS ONE